# Uniform Manifold Approximation with Two-phase Optimization

## Abstract

We present a dimensionality reduction algorithm called Uniform Manifold Approximation with Two-phase Optimization (UMATO) which produces less biased global structures in the embedding results and is robust over diverse initialization methods than previous methods such as $t$-SNE and UMAP. We divide the optimization into two phases to alleviate the bias by establishing the global structure early using the representatives of the high-dimensional structures. The phases are 1) global optimization to obtain the overall skeleton of data and 2) local optimization to identify the regional characteristics of local areas. In our experiments with one synthetic and three real-world datasets, UMATO outperformed widely-used baseline algorithms, such as PCA, Isomap, $t$-SNE, UMAP, topological autoencoders and Anchor $t$-SNE, in terms of quality metrics and 2D projection results.

## 1 Introduction

We present a novel dimensionality reduction method, Uniform Manifold Approximation with Two-phase Optimization (UMATO) to obtain less biased and robust embedding over diverse initialization methods. One effective way of understanding high-dimensional data in various domains is to reduce its dimensionality and investigate the projection in a lower-dimensional space. The limitation of previous approaches such as $t$-Stochastic Neighbor Embedding ($t$-SNE, Maaten & Hinton (2008)) and Uniform Manifold Approximation and Projection (UMAP, McInnes et al. (2018)) is that they are susceptible to different initialization methods, generating considerably different embedding results (Section 5.5).

$t$-SNE adopts Kullback-Leibler (KL) divergence as its loss function. The fundamental limitation of the KL divergence is that the penalty for the points that are distant in the original space being close in the projected space is too little (Appendix B). This results in only the local manifolds being captured, while clusters that are far apart change their relative locations from run to run. Meanwhile, UMAP leverages the cross-entropy loss function, which is known to charge a penalty for points that are distant in the original space being close in the projection space and for points that are close in the original space being distant in the projection space (Appendix B). UMAP considers all points in the optimization at once with diverse sampling techniques (i.e., negative sampling and edge sampling). Although the approximation technique in UMAP optimization makes the computation much faster, this raises another problem that the clusters in the embedding become dispersed as the number of epochs increases (Appendix K), which can lead to misinterpretation. UMAP tried to alleviate this by using a fixed number (e.g., 200), which is ad hoc, and by applying a learning rate decay. However, the optimal number of epochs and decay schedule for each initialization method needs to be found in practice.

To solve the aforementioned problems, we avoid using approximation during the optimization process, which normally would result in greatly increased computational cost. Instead, we first run optimization only with a small number of points that represent the data (i.e., hub points). Finding the optimal projection for a small number of points using a cross-entropy function is relatively easy and robust, making the additional techniques employed in UMAP unnecessary. Furthermore, it is less sensitive to the initialization method used (Section 5.5). After capturing the overall skeleton of the high-dimensional structure, we gradually append the rest of the points in subsequent phases. Although the same approximation technique as UMAP is used for these points, as we have already embedded the hub points and use them as anchors, the projections become more robust and unbiased. The gradual addition of points can in fact be done in a single phase; we found additional phases

do not result in meaningful improvements in the performance but only in the increased computation time (Section 4.5). Therefore, we used only two phases in UMAP: global optimization to capture the global structures (i.e., the pairwise distances in a high-dimensional space) and local optimization to retain the local structures (i.e., the relationships between neighboring points in a high-dimensional space) of the data.

We compared UMATO with popular dimensionality reduction techniques including PCA, Isomap (Tenenbaum et al. (2000)), $t$-SNE, UMAP, topological autoencoders (Moor et al. (2020)) and A$t$-SNE (Fu et al. (2019)). We used one synthetic (101-dimensional spheres) and three real-world (MNIST, Fashion MNIST, and Kuzushiji MNIST) datasets and analyzed the projection results with several quality metrics. In conclusion, UMATO demonstrated better performance than the baseline techniques in all datasets in terms of $KL_\sigma$ with different $\sigma$ values, meaning that it reasonably preserved the density of data over diverse length scales. Finally, we presented the 2D projections of each dataset, including the replication of an experiment using the synthetic Spheres dataset introduced by Moor et al. (2020) where data points locally constitute multiple small balls globally contained in a larger sphere. Here, we demonstrate that UMATO can better preserve both structures compared to the baseline algorithms (Figure 3).

## 2    RELATED WORK

**Dimensionality reduction.**  Most previous dimensionality reduction algorithms focused on preserving the data's local structures. For example, Maaten & Hinton (2008) proposed $t$-SNE, focusing on the crowding problem with which the previous attempts (Hinton & Roweis (2002); Cook et al. (2007)) have struggled, to visualize high-dimensional data through projection produced by performing stochastic gradient descent on the KL divergence between two density functions in the original and projection spaces. Van Der Maaten (2014) accelerated $t$-SNE developing a variant of the Barnes-Hut algorithm (Barnes & Hut (1986)) and reduced the computational complexity from $O(N^2)$ into $O(N \log N)$. After that, grounded in Riemannian geometry and algebraic topology, McInnes et al. (2018) introduced UMAP as an alternative to $t$-SNE. Leveraging the cross-entropy function as its loss function, UMAP reduced the computation time by employing negative sampling from Word2Vec (Mikolov et al. (2013)) and edge sampling from LargeVis (Tang et al. (2015; 2016)) (Table 1). Moreover, they showed that UMAP can generate stable projection results compared to $t$-SNE over repetition.

On the other hand, there also exist algorithms that aim to capture the global structures of data. Isomap (Tenenbaum et al. (2000)) was proposed to approximate the geodesic distance of high-dimensional data and embed it onto the lower dimension. Global $t$-SNE (Zhou & Sharpee (2018)) converted the joint probability distribution, $P$, in the high-dimensional space from Gaussian to Student's-$t$ distribution, and proposed a variant of KL divergence. By adding it with the original loss function of $t$-SNE, Global $t$-SNE assigns a relatively large penalty for a pair of distant data points in high-dimensional space being close in the projection space. Another example is topological autoencoders (Moor et al. (2020)), a deep-learning approach that uses a generative model to make the latent space resemble the high-dimensional space by appending a topological loss to the original reconstruction loss of autoencoders. However, they required a huge amount of time for hyperparameter exploration and training for a dataset, and only focused on the global aspect of data. Unlike other techniques that presented a variation of loss functions in a single pipeline, UMATO is novel as it preserves both structures by dividing the optimization into two phases; this makes it outperform the baselines with respect to quality metrics in our experiments.

**Hubs, landmarks, and anchors.**  Many dimensionality reduction techniques have tried to draw sample points to better model the original space; these points are usually called hubs, landmarks, or anchors. Silva & Tenenbaum (2003) proposed Landmark Isomap, a landmark version of classical multidimensional scaling (MDS) to alleviate its computation cost. Based on the Landmark Isomap, Yan et al. (2018) tried to retain the topological structures (i.e., homology) of high-dimensional data by approximating the geodesic distances of all data points. However, both techniques have the limitation that landmarks were chosen randomly without considering their importance. UMATO uses a $k$-nearest neighbor graph to extract significant hubs that can represent the overall skeleton of high-dimensional data. The most similar work to ours is A$t$-SNE (Fu et al. (2019)), which optimized the anchor points and all other points with two different loss functions. However, since the anchors wander during the optimization and the KL divergence does not care about distant points, it hardly

captures the global structure. UMATO separates the optimization process into two phases so that the hubs barely moves but guides other points so that the subareas manifest the shape of the high-dimensional manifold in the projection. Applying different cross-entropy functions to each phase also helps preserve both structures.

## 3 UMAP

Since UMATO shares the overall pipeline of UMAP (McInnes et al. (2018)), we briefly introduce UMAP in this section. Although UMAP is grounded in a sophisticated mathematical foundation, its computation can be simply divided into two steps, graph construction and layout optimization, a configuration similar to $t$-SNE. In this section, we succinctly explain the computation in an abstract manner. For more details about UMAP, please consult the original paper (McInnes et al. (2018)).

**Graph Construction.** UMAP starts by generating a weighted $k$-nearest neighbor graph that represents the distances between data points in the high-dimensional space. Given an input dataset $X = \{x_1, \ldots, x_n\}$, the number of neighbors to consider $k$ and a distance metric $d : X \times X \to [0, \infty)$, UMAP first computes $\mathcal{N}_i$, the $k$-nearest neighbors of $x_i$ with respect to $d$. Then, UMAP computes two parameters, $\rho_i$ and $\sigma_i$, for each data point $x_i$ to identify its local metric space. $\rho_i$ is a nonzero distance from $x_i$ to its nearest neighbor:

$$\rho_i = \min_{j \in \mathcal{N}_i} \{d(x_i, x_j) \mid d(x_i, x_j) > 0\}. \tag{1}$$

Using binary search, UMAP finds $\sigma_i$ that satisfies:

$$\sum_{j \in \mathcal{N}_i} \exp(-\max(0, d(x_i, x_j) - \rho_i)/\sigma_i) = \log_2(k). \tag{2}$$

Next, UMAP computes:

$$v_{j|i} = \exp(-\max(0, d(x_i, x_j) - \rho_i)/\sigma_i), \tag{3}$$

the weight of the edge from a point $x_i$ to another point $x_j$. To make it symmetric, UMAP computes $v_{ij} = v_{j|i} + v_{i|j} - v_{j|i} \cdot v_{i|j}$, a single edge with combined weight using $v_{j|i}$ and $v_{i|j}$. Note that $v_{ij}$ indicates the similarity between points $x_i$ and $x_j$ in the original space. Let $y_i$ be the projection of $x_i$ in a low-dimensional projection space. The similarity between two projected points $y_i$ and $y_j$ is $w_{ij} = (1 + a||y_i - y_j||_2^{2b})^{-1}$, where $a$ and $b$ are positive constants defined by the user. Setting both $a$ and $b$ to 1 is identical to using Student's $t$-distribution to measure the similarity between two points in the projection space as in $t$-SNE (Maaten & Hinton (2008)).

**Layout Optimization.** The goal of layout optimization is to find the $y_i$ that minimizes the difference (or loss) between $v_{ij}$ and $w_{ij}$. Unlike $t$-SNE, UMAP employs the cross entropy:

$$C_{UMAP} = \sum_{i \neq j} [v_{ij} \cdot \log(v_{ij}/w_{ij}) - (1 - v_{ij}) \cdot \log((1 - v_{ij})/(1 - w_{ij}))], \tag{4}$$

between $v_{ij}$ and $w_{ij}$ as the loss function. UMAP initializes $y_i$ through spectral embedding (Belkin & Niyogi (2002)) and iteratively optimize its position to minimize $C_{UMAP}$. Given the output weight $w_{ij}$ as $1/(1 + ad_{ij}^{2b})$, the attractive gradient is:

$$\frac{C_{UMAP}^+}{y_i} = \frac{-2abd_{ij}^{2(b-1)}}{1 + ad_{ij}^{2b}} v_{ij}(y_i - y_j), \tag{5}$$

and the repulsive gradient is:

$$\frac{C_{UMAP}^-}{y_i} = \frac{2b}{(\epsilon + d_{ij}^2)(1 + ad_{ij}^{2b})}(1 - v_{ij})(y_i - y_j), \tag{6}$$

where $\epsilon$ is a small value added to prevent division by zero and $d_{ij}$ is a Euclidean distance between $y_i$ and $y_j$. For efficient optimization, UMAP leverages the negative sampling technique from Word2Vec (Mikolov et al. (2013)). After choosing a target point and its negative samples, the position of the target is updated with the attractive gradient, while the positions of the latter do so with

the repulsive gradient. Moreover, UMAP utilizes edge sampling (Tang et al. (2015; 2016)) to accelerate and simplify the optimization process (Table 1). In other words, UMAP randomly samples edges with a probability proportional to their weights, and subsequently treats the selected ones as binary edges. Considering the previous sampling techniques, the modified objective function is:

$$O = \sum_{(i,j) \in E} v_{ij}(\log(w_{ij}) + \sum_{k=1}^{M} E_{j_k \sim P_n(j)} \gamma \log(1 - w_{ij_k})). \quad (7)$$

Here, $v_{ij}$ and $w_{ij}$ are the similarities in the high and low-dimensional spaces respectively, $M$ is the number of negative samples and $E_{j_k \sim P_n(j)}$ indicates that $j_k$ is sampled according to a noisy distribution, $P_n(j)$, from Word2Vec (Mikolov et al. (2013)).

## 4 UMATO

Figure 2 illustrates the computation pipeline of UMATO, which delineates the two-phase optimization (see Figure 9 for a detailed illustration of the overall pipeline). As a novel approach, we split the optimization into global and local so that it could generate a low-dimensional projection keeping both structures well-maintained. We present the pseudocode of UMATO in Appendix A, and made the source codes of it publicly available[1].

### 4.1 POINTS CLASSIFICATION

In the big picture, UMATO follows the pipeline of UMAP. We first find the $k$-nearest neighbors in the same way as UMAP, by assuming the local connectivity constraint, i.e., no single point is isolated and each point is connected to at least a user-defined number of points. After calculating $\rho$ (Equation 1) and $\sigma$ (Equation 2) for each point, we obtain the pairwise similarity for every pair of points. Once the $k$-nearest neighbor indices are established, we unfold it and check the frequency of each point to sort them into descending order so that the index of the popular points come to the front.

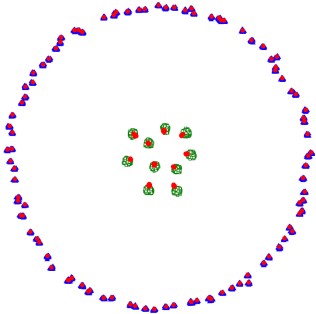

Figure 1: **Points classification using Spheres dataset.** Each point is classified into a hub (red circles), an expanded nearest neighbor (green squares), or an outlier (blue triangles). Best viewed in color.

Then, we build a $k$-nearest neighbor graph by repeating the following steps until no points remain unconnected: 1) choose the most frequent point as a hub among points that are not already connected, 2) retrieve the $k$-nearest neighbors of the chosen point (i.e., hub), and the points selected from steps 1 and 2 will become a connected component. The gist is that we divide the points into three disjoint sets: hubs, expanded nearest neighbors, and outliers (Figure 1). Thanks to the sorted indices, the most popular point in each iteration—but not too densely located—becomes the hub point. Once the hub points are determined, we recursively seek out their nearest neighbors and again look for the nearest neighbors of those neighbors, until there are no points to be newly appended. In other words, we find all connected points that are expanded from the original hub points, which, in turn, is called the expanded nearest neighbors. Any remaining point that is neither a hub point nor a part of any expanded nearest neighbors is classified as an outlier. The main reason to rule out the outliers is, similar to the previous approach (Gong et al. (2012)), to achieve the robustness of the practical manifold learning algorithm. As the characteristics of these classes differ significantly, we take a different approach for each class of points to obtain both structures. That is, we run global optimization for the hub points (Section 4.2), local optimization for the expanded nearest neighbors (Section 4.3), and no optimization for the outliers (Section 4.4). In the next section we explain each in detail.

### 4.2 GLOBAL OPTIMIZATION

After identifying hub points, we run the global optimization to retrieve the skeletal layout of the data. First, we initialize the positions of hub points using PCA, which makes the optimization process

---

[1]https://www.github.com/anonymous-author/anonymous-repo

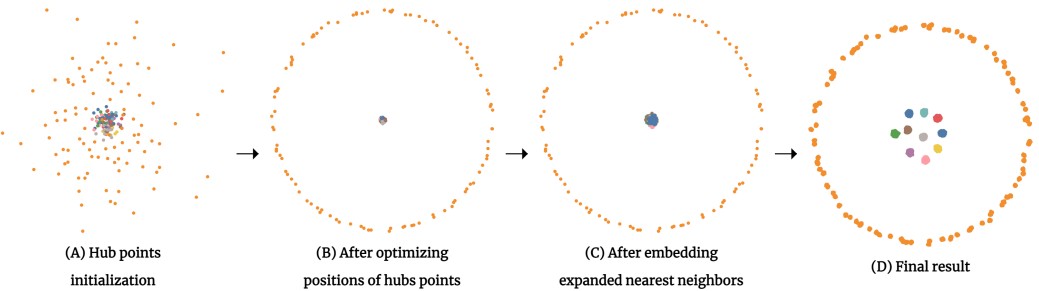

| (A) Hub points | (B) After optimizing | (C) After embedding | (D) Final result |
| initialization | positions of hubs points | expanded nearest neighbors | |

Figure 2: **An illustration of UMATO pipeline using 10,000 data points (101 dimensions) of the Spheres.** (A) UMATO first initializes hub points using PCA, (B) then optimize their positions using the cross entropy function. (C) Next, we embed the expanded nearest neighbors to the projection and optimize their positions using sampling techniques for acceleration. (D) Lastly, we append the outliers and achieve the final projection result. Best viewed in color.

faster and more stable than using random initial positions. Next, we optimize the positions of hub points by minimizing the cross-entropy function (Equation 4). Let $f(X) = \{f(x_i, x_j)|x_i, x_j \in X\}$ and $g(Y) = \{g(y_i, y_j)|y_i, y_j \in Y\}$ be two adjacency matrices in high- and low-dimensional spaces. If $X_h$ represents a set of points selected as hubs in high-dimensional space, and $Y_h$ is a set of corresponding points in the projection, we minimize the cross entropy—$CE(f(X_h)||g(Y_h))$—between $f(X_h)$ and $g(Y_h)$.

UMAP computes the cross-entropy between all existing points using two sampling techniques, edge sampling and negative sampling, for speed (Table 1). However, this often ends up capturing only the local properties of data because of the sampling biases and thus it cannot be used for cases that require a comprehensive understanding of the data. On the other hand, in its first phase, UMATO only optimizes for representatives (i.e., the hub points) of data, which takes much less time but can still approximate the manifold effectively.

Table 1: **The runtime for each algorithm using MNIST dataset.** UMAP and UMATO take much less time than MulticoreT-SNE (Ulyanov (2016)) when tested on a Linux server with 40-core Intel Xeon Silver 4210 CPUs. The runtimes are averaged over 10 runs. Isomap (Tenenbaum et al. (2000)) took more than 3 hours to get the embedding result.

| Algorithm | Runtime (s) |
|-----------|-------------|
| Isomap | 3 hours > |
| $t$-SNE | $374.85 \pm 11.38$ |
| UMAP | $26.10 \pm 3.97$ |
| UMATO | $73.32 \pm 8.39$ |

### 4.3 LOCAL OPTIMIZATION

In the second phase, UMATO embeds the expanded nearest neighbors to the projection that is computed using only the hub points from the first phase. For each point in the expanded nearest neighbors, we retrieve its nearest $m$ (e.g., 10) hubs in the original high-dimensional space and set its initial position in the projection to the average positions of the hubs in the projection with small random perturbations. We follow a similar optimization process as UMAP in the local optimization with small differences. As explained in Section 3, UMAP first constructs the graph structure; we perform the same task but only with the hubs and expanded nearest neighbors. While doing this, since some points are excluded as outliers, we need to update the $k$-nearest neighbor indices. This is fast because we recycle the already-built $k$-nearest neighbor indices by updating the outliers to the new nearest neighbor.

Once we compute the similarity between points (Equation 3), to optimize the positions of points, similar to UMAP, we use the cross-entropy loss function with edge sampling and negative sampling (Equation 7). Here, we try to avoid moving the hubs as much as possible since they have already formed the global structure. Thus, we only sample a point $p$ among the expanded nearest neighbors as one end of an edge, while the point $q$ at the other end of the edge can be chosen from all points except outliers. In UMAP implementation, when $q$ pulls in $p$, $p$ also drags $q$ to facilitate the optimization (Equation 5). When updating the position of $q$, we only give a penalty to this (e.g., 0.1), if $q$ is a hub point, not letting its position excessively be affected by $p$. In addition, because the repulsive force can disperse the local attachment, making the point veer off for each epoch and eventually destroying the well-shaped global layout, we multiply a penalty (e.g., 0.1) when calculating the repulsive gradient (Equation 6) for the points selected as negative samples.

### 4.4 OUTLIERS ARRANGEMENT

Since the isolated points, which we call outliers, mostly have the same distance to all the other data points in high-dimensional space, due to the curse of dimensionality, they both sabotage the global structure we have already made and try to mingle with all other points, thus distorting the overall projection. We do not optimize these points but instead simply append them using the already-projected points (e.g., hubs or expanded nearest neighbors), that belong to each outlier's connected component of the nearest neighbor graph. That is, if $x_i \in C_n$ where $x_i$ is the target outlier and $C_n$ is the connected component to which $x_i$ belongs, we find $x_j \in C_n$ that has already been projected and is closest to $x_i$. We arrange $y_i$ which corresponds to $x_i$ in low-dimensional space using the position of $y_j$ in the same component offset by a random noise. In this way, we can benefit from the comprehensive composition of the projection that we have already optimized when arranging the outliers. We can ensure that all outliers can find a point as its neighbor since we picked hubs from each connected component of the nearest neighbor graph and thus at least one point is already located and has an optimized position (Section 4.2).

### 4.5 MULTI-PHASE OPTIMIZATION

The optimization of UMATO can be easily expanded to multiple phases (e.g., three or more phases). Since we have a recursive procedure to expand the nearest neighbors, we can insert the optimization process each time we expand the neighbors to create a multi-phase algorithm. However, our experiment with three- and four-phase optimization with the Fashion MNIST dataset showed that there is no big difference between two-phase optimization and that with more than two phases. Appendix C contains the quantitative and qualitative results of the experiment for multi-phase optimization.

## 5 EXPERIMENTS

We conducted experiments to evaluate UMATO's ability to capture the global and local structures of high-dimensional data. We compared UMATO with six baseline algorithms, PCA, Isomap, $t$-SNE, UMAP, topological autoencoders, and A$t$-SNE in terms of global (i.e., DTM and KL$_\sigma$) and local (i.e., trustworthiness, continuity, and MRREs) quality metrics.

### 5.1 DATASETS

We used four datasets for the experiments: Spheres, MNIST (LeCun & Cortes (2010)), Fashion MNIST (Xiao et al. (2017)), and Kuzushiji MNIST (Clanuwat et al. (2018)). Spheres is a synthetic dataset that has 10,000 rows of 101 dimensions. It has a high-dimensional structure in which ten small spheres are contained in a larger sphere. Specifically, the dataset's first 5,000 rows are the points sampled from a sphere of radius 25 and 500 points are sampled for each of the ten smaller spheres of radius 5 shifted to a random direction from the origin. This dataset is the one used for the original experiment with topological autoencoders (Moor et al. (2020)). Other datasets are images of digits, fashion items, and Japanese characters, each of which consists of 60,000 784-dimensional ($28 \times 28$) images from 10 classes.

### 5.2 EXPERIMENTAL SETTING

**Evaluation Metrics.** To assess how well projections preserve the global structures of high-dimensional data, we computed the density estimates (Chazal et al. (2011; 2017)), the so-called Distance To a Measure (DTM), between the original data and the projections. Moor et al. (2020) adopted the Kullback-Leibler divergence between density estimates with different scales (KL$_\sigma$) to evaluate the global structure preservation. To follow the original experimental setup by Moor et al. (2020), we found the projections with the lowest KL$_{0.1}$ from all algorithms by adjusting their hyperparameters. Next, to evaluate the local structure preservation of projections, we used the mean relative rank errors (MRREs, Lee & Verleysen (2007)), trustworthiness, and continuity (Venna & Kaski (2001)). All of these local quality metrics estimate how well the nearest neighbors in one space (e.g., high- or low-dimensional space) are preserved in the other space. For more information on the quality metrics, we refer readers to Appendix E.

**Baselines.** We set the most widely used dimensionality reduction techniques as our baselines, including PCA, Isomap (Tenenbaum et al. (2000)), $t$-SNE (Maaten & Hinton (2008)),

Table 2: **Quantitative results of UMATO and six baseline algorithms.** The hyperparameters of the algorithms are chosen to minimize $KL_{0.1}$. The best one is in bold and underlined, and the runner-up is in bold. Only first four digits are shown for conciseness.

| Dataset | Method | Global quality metrics | | | | Local quality metrics | | | |
|---|---|---|---|---|---|---|---|---|---|
| | | DTM | $KL_{0.01}$ | $KL_{0.1}$ | $KL_1$ | Cont | Trust | $MRRE_X$ | $MRRE_Z$ |
| Spheres | PCA | 0.9950 | 0.7568 | 0.6525 | 0.0153 | 0.7983 | 0.6088 | 0.7985 | 0.6078 |
| | Isomap | 0.7784 | 0.4492 | 0.4267 | 0.0095 | **0.9041** | 0.6266 | **0.9039** | 0.6268 |
| | t-SNE | 0.9116 | 0.6070 | 0.5365 | 0.0128 | **0.8903** | **_0.7073_** | **0.9032** | **_0.7261_** |
| | UMAP | 0.9209 | 0.6100 | 0.5383 | 0.0134 | 0.8760 | 0.6499 | 0.8805 | 0.6494 |
| | TopoAE | **0.6890** | **0.2063** | **0.3340** | **0.0076** | 0.8317 | 0.6339 | 0.8317 | 0.6326 |
| | At-SNE | 0.9448 | 0.6584 | 0.5712 | 0.0138 | 0.8721 | 0.6433 | 0.8768 | 0.6424 |
| | UMATO | **_0.3888_** | **_0.1341_** | **_0.1434_** | **_0.0014_** | 0.7884 | **0.6558** | 0.7887 | **0.6557** |
| Fashion MNIST | PCA | 0.2315 | 0.6929 | 0.0454 | **_0.0006_** | 0.9843 | 0.9117 | 0.9853 | 0.9115 |
| | Isomap | **0.2272** | **_0.6668_** | **0.0446** | 0.0010 | 0.9865 | 0.9195 | 0.9872 | 0.9196 |
| | t-SNE | 0.2768 | 0.8079 | 0.0663 | 0.0017 | 0.9899 | **_0.9949_** | 0.9919 | **_0.9955_** |
| | UMAP | 0.2755 | 0.8396 | 0.0641 | 0.0016 | **_0.9950_** | 0.9584 | **_0.9955_** | 0.9584 |
| | TopoAE | 0.2329 | 0.7301 | 0.0446 | **0.0008** | 0.9908 | 0.9591 | 0.9913 | 0.9590 |
| | At-SNE | 0.2973 | 0.8389 | 0.0702 | 0.0017 | 0.9826 | **0.9847** | 0.9849 | **0.9848** |
| | UMATO | **_0.2035_** | 0.6852 | **_0.0342_** | 0.0008 | **0.9911** | 0.9500 | **0.9919** | 0.9502 |
| MNIST | PCA | 0.4104 | 1.4981 | 0.1349 | 0.0014 | 0.9573 | 0.7340 | 0.9605 | 0.7342 |
| | Isomap | **_0.3358_** | **_1.0361_** | **_0.0857_** | **0.0012** | 0.9743 | 0.7527 | 0.976 | 0.7528 |
| | t-SNE | 0.4263 | 1.4964 | 0.1523 | 0.0024 | **0.9833** | **_0.9954_** | 0.9869 | **_0.9963_** |
| | UMAP | 0.4172 | 1.5734 | 0.1430 | 0.0026 | **0.9891** | 0.9547 | **0.9907** | 0.9547 |
| | TopoAE | 0.3686 | 1.3818 | 0.1048 | **_0.0011_** | 0.9716 | 0.9429 | 0.9732 | 0.9429 |
| | At-SNE | 0.4328 | 1.5623 | 0.1482 | 0.0018 | 0.9768 | **0.9765** | 0.9830 | **0.9777** |
| | UMATO | **0.3525** | **1.2785** | **0.1017** | 0.0014 | 0.9792 | 0.8421 | 0.9813 | 0.8422 |
| Kuzushiji MNIST | PCA | 0.4215 | 0.1710 | 0.1317 | 0.0014 | 0.9380 | 0.7213 | 0.9420 | 0.7211 |
| | Isomap | **0.3458** | 0.2171 | **0.0906** | **0.0012** | 0.9573 | 0.7638 | 0.9589 | 0.7635 |
| | t-SNE | 0.4254 | **0.0483** | 0.1369 | 0.0025 | 0.9843 | **_0.9688_** | 0.9871 | **_0.9693_** |
| | UMAP | 0.3873 | **0.0417** | 0.1148 | 0.0026 | **0.9893** | 0.9563 | **0.9908** | 0.9564 |
| | TopoAE | 0.3730 | 0.1495 | 0.1027 | **_0.0011_** | 0.9755 | 0.9442 | 0.9768 | 0.9440 |
| | At-SNE | **0.3505** | 0.0807 | **0.0978** | **0.0013** | 0.9786 | **0.9671** | 0.9824 | **0.9676** |
| | UMATO | **_0.3231_** | 0.1365 | **_0.0815_** | 0.0016 | **0.9865** | 0.8888 | **0.9881** | 0.8895 |

UMAP (McInnes et al. (2018)), and At-SNE (Fu et al. (2019)). In the case of t-SNE, we leveraged Multicore t-SNE (Ulyanov (2016)) for fast computation. To initialize the points' position, we used PCA for t-SNE, following the recommendation in the previous work (Linderman et al. (2019)), and spectral embedding for UMAP which was set to default. In addition, we compared with topological autoencoders (Moor et al. (2020)) that were developed to capture the global properties of the data using a deep learning-based generative model. Following the convention of visualization in dimensionality reduction, we determined our result projected onto 2D space. We tuned the hyperparameters of each technique to minimize the $KL_{0.1}$. Appendix F further describes the details of the hyperparameters settings.

## 5.3 QUANTITATIVE RESULTS

Table 2 displays the experiment results. In most cases, UMATO was the only method that has shown performance both in the global and local quality metrics in most datasets. For local metrics, t-SNE, At-SNE, and UMAP generally had the upper-hand, but UMATO showed comparable $MRRE_X$ and continuity in Spheres, Fashion MNIST, and Kuzushiji MNIST datasets. Meanwhile, Isomap and topological autoencoders were mostly good at global quality metrics, although UMATO had the lowest (best) $KL_{0.1}$ and DTM except for the MNIST dataset.

## 5.4 QUALITATIVE RESULTS

Among the five algorithms, only UMATO could preserve both the global and local structure of the Spheres dataset. If we look at the figure made by UMATO, the outer sphere encircles the inner spheres in a circular form, which is the most intuitive to understand the relationship among different classes and the local linkage in detail. In the results from Isomap, t-SNE, UMAP, and At-SNE,

Figure 3: **2D projections produced by UMATO and six baseline algorithms.** $t$-SNE, A$t$-SNE, and UMAP showed as if the points from a surrounding sphere were attached to inner spheres, not reflecting the data's global structures. PCA, Isomap and topological autoencoders attempted to preserve the global structures, but failed to manifest the complicated hierarchical structures. UMATO was the only algorithm to capture both the global and local structures among all different sphere classes; this is best viewed in color.

the points representing the surrounding giant sphere mix with those representing the other small inner spheres, thus failing to capture the nested relationships among different classes. Meanwhile, topological autoencoders are able to realize the global relationship between classes in an incomplete manner; the points for the outer sphere are too spread out, thus losing the local characteristics of the class. From this result, we can acknowledge how UMATO can work with high-dimensional data effectively to reveal both global and local structures. 2D visualization results on other datasets (MNIST, Fashion MNIST, Kuzushiji MNIST) can be found in Appendix H. Lastly, we report an additional experiment on the mouse neocortex dataset (Tasic et al. (2018)) in Appendix N which shows the relationship between classes much better than the baseline algorithms like $t$-SNE and UMAP.

## 5.5 PROJECTION ROBUSTNESS OVER DIVERSE INITIALIZATION METHODS

We experimented with the robustness of each dimensionality reduction technique with different initialization methods such as PCA, spectral embedding, random position, and class-wise separation. In class-wise separation, we initialized each class with a non-overlapping random position in 2-dimensional space, adding random Gaussian noise. In our results, UMATO embeddings were almost the same on the real-world datasets, while the UMAP and t-SNE results relied highly upon the initialization method. We report this in Table 3 with a quantitative comparison using Procrustes distance. Specifically, given two datasets $X = \{x_1, x_2, \ldots, x_n\}$ and $Y = \{y'_1, y'_2, \ldots, y'_n\}$ where $y'_i$ corresponds to $x_i$, the Procrustes distance is defined as

$$d_P(X, Y) = \sqrt{\sum_{i=1}^{N} (x_i - y'_i)}. \tag{8}$$

For all cases, we ran optimal translation, uniform scaling, and rotation to minimize the Procrustes distance between the two distributions. In the case of the Spheres dataset, as defined in Appendix G, the clusters were equidistant from each other. The embedding results have to be different due to the limitation of the 2-dimensional space since there is no way to express this relationship. However, as we report in Figure 4, the global and local structures of the Spheres data are manifested with UMATO with all different initialization methods.

Table 3: **The average value of normalized Procrustes distance between diverse dimensionality reduction techniques over four datasets.** In all real-world datasets, UMATO has shown the most robust embedding results over different initialization methods. Although the UMATO results in the highest normalized Procrustes distance in the Spheres dataset, the embedding results look quite similar (Figure 4). The winner is in bold.

| Sample (%) | Spheres | MNIST | FMNIST | KMNIST |
|---|---|---|---|---|
| $t$-SNE | 0.7878 | 0.8665 | 0.8284 | 0.8668 |
| UMAP | **0.7726** | 0.7767 | 0.7793 | 0.8213 |
| UMATO | 0.9504 | **0.4808** | **0.0120** | **0.2037** |

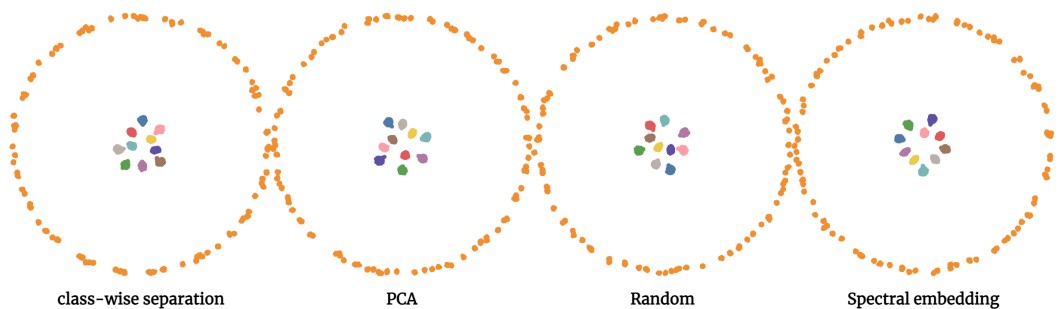

Figure 4: **UMATO results on the Spheres dataset using different initialization methods.** Although the average value of the normalized Procrustes distance of UMATO results is higher than the baselines because of the equidistant clusters of inner spheres, both global and local structures are well-captured with all different initialization methods. Best viewed in color.

## 6 CONCLUSION

We present a two-phase dimensionality reduction algorithm called UMATO that can effectively preserve the global and local properties of high-dimensional data. In our experiments with diverse datasets, we have proven that UMATO can outperform previous widely used baselines (e.g., $t$-SNE and UMAP) both quantitatively and qualitatively. As future work, we plan to accelerate UMATO, as in previous attempts with other dimensionality reduction techniques (Pezzotti et al. (2019); Nolet et al. (2020)), by implementing it on a heterogeneous system (e.g., GPU) for speedups.

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

## A  UMATO Algorithm Pseudocode

The pseudocode of UMATO is as below:

---
**Algorithm 1** Uniform Manifold Approximation with Two-phase Optimization

---
1:  **procedure** UMATO($X, k_X, d, min\_dist, n_h, e_g, e_l$)
**Input:** High-dimensional data $X$, number of nearest neighbors $k$, projection dimension $d$, minimum distance in projection result $min\_dist$, number of hub points $n_h$, epochs for global and local optimization $e_g, e_l$
**Output:** Low-dimensional projection $Y$
2:      Compute $k$-nearest neighbors of $X$
3:      Obtain sorted list using indices' frequency of $k$-nearest neighbors
4:      Build $k$-nearest neighbor graph structure
5:      Classify points into hubs, expanded nearest neighbors, and outliers
6:      Optimize $CE(f(X_h)||g(Y_h))$ to preserve global configuration (Equation 4)
7:      Initialize expanded nearest neighbors using hub locations
8:      Update $k$-nearest neighbors & compute weights (Equation 3)
9:      Optimize $CE(f(X)||g(Y))$ to preserve local configuration (Equation 7)
10:     Position outliers
11:     **return** $Y$
12: **end procedure**

---

## B  The Meaning of Using Different Loss Functions in Dimensionality Reduction

Following the notations from above, we set the similarity between points in high-dimensional space $x_i$ and $x_j$ as $v_{ij}$ and the low-dimensional space $y_i$ and $y_j$ as $w_{ij}$. Then we can write the KL divergence and cross entropy loss function as:

$$KL = \sum_{i \neq j} v_{ij} \cdot \log(v_{ij}/w_{ij}), \tag{9}$$

$$CE = \sum_{i \neq j} v_{ij} \cdot \log(v_{ij}/w_{ij}) - \sum_{i \neq j}(1 - v_{ij}) \cdot \log((1 - v_{ij})/(1 - w_{ij})). \tag{10}$$

Table 4: **Analysis of the KL divergence and cross-entropy loss function for imposing penalties when updating the positions of points in low-dimensional space.** (Upper table) The KL divergence and the first term of the cross-entropy function impose a big penalty when $w_{ij}$ is small but $v_{ij}$ is large. (Lower table) In contrast, the second term of cross-entropy function imposes a big penalty when $v_{ij}$ is small but $w_{ij}$ is large.

| | | Low $d$ | |
| | | Large $w_{ij}$ | Small $w_{ij}$ |
|---|---|---|---|
| High $d$ | Large $v_{ij}$ | a. Small penalty | b. Big penalty (preserves local structures) |
| | Small $v_{ij}$ | c. Small penalty (ignores global structures) | d. Small penalty |

| | | Low $d$ | |
| | | Large $w_{ij}$ | Small $w_{ij}$ |
|---|---|---|---|
| High $d$ | Large $v_{ij}$ | e. Small penalty | f. Small penalty (ignores local structures) |
| | Small $v_{ij}$ | g. Big penalty (preserves global structures) | h. Small penalty |

If we use the same probability distributions, $v_{ij}$ and $w_{ij}$, the KL divergence and the first term of cross-entropy are exactly the same. If $v_{ij}$ and $w_{ij}$ are both large (Table 4 a.) or both small (Table 4 d.), this means that the relationship between points in high-dimensional space is well-retained in the projection. Thus, the positions of points in the low-dimensional space do not have to move. As $v_{ij}$ and $w_{ij}$ are similar, $\log(v_{ij}/w_{ij})$ becomes zero, producing a small cost in the end.

However, we need to modify the position of $w_{ij}$ if there exists a gap between $v_{ij}$ and $w_{ij}$. The KL divergence imposes a big penalty when $v_{ij}$ is large but $w_{ij}$ is small (Table 4 b.). That is, if the neighboring points in high-dimensional space are not captured well in the projection, the KL divergence imposes a high penalty to move the point ($v_{ij}$) into the right position. Thus, we can understand why $t$-SNE is able to capture the local characteristics of high-dimensional space, but not the global ones. However, the second term of cross-entropy imposes a big penalty when $v_{ij}$ is small but $w_{ij}$ is large (Table 4 g.). Therefore, it moves points that are close together in the high-dimensional space but far apart in the projection.

## C  MULTI-PHASE OPTIMIZATION

We report the result of multi-phase optimization (e.g., three and four-phase) using the Fashion MNIST dataset both quantitatively (Table 5) and qualitatively (Figure 5). As in Figure 5, we were unable to find any significant differences between the 2D projections, although some outliers were located in different places. Moreover, the quality metrics were almost the same for all three results. The original UMATO was the winner in DTM, $KL_{0.1}$, $KL_1$, continuity, and $MRRE_Z$ but came last in other quality metrics. In addition, the gap in metrics between UMATO and the multi-phase optimizations indicated a trivial difference. Thus, we concluded that developing a multi-phase optimization for UMATO does not bring about any notable improvement in the projection result.

Table 5: **Quantitative evaluation of UMATO and UMATO with multi-phase optimizations.** Although the optimization process of UMATO can be simply expandable for multiple phases, no apparent distinctions are found in the the results with different numbers of optimization phases. The winner is in bold.

| Dataset | Method | DTM | $KL_{0.01}$ | $KL_{0.1}$ | $KL_1$ | Cont | Trust | $MRRE_X$ | $MRRE_Z$ |
|---|---|---|---|---|---|---|---|---|---|
| Fashion MNIST | UMATO | **0.2035** | 0.6852 | **0.0342** | **0.0008** | **0.9911** | 0.9500 | **0.9919** | 0.9502 |
| | 3-Phases | 0.2058 | 0.6546 | 0.0343 | **0.0008** | 0.9900 | **0.9556** | 0.9909 | **0.9561** |
| | 4-Phases | 0.2095 | **0.6533** | 0.0359 | **0.0008** | 0.9895 | 0.9532 | 0.9904 | 0.9536 |

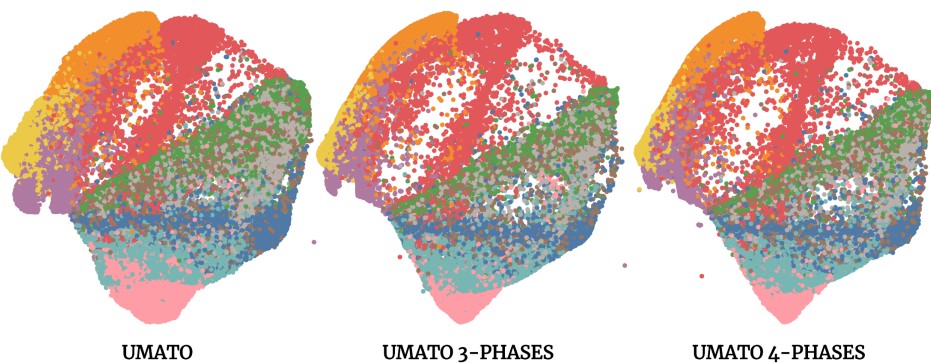

UMATO          UMATO 3-PHASES          UMATO 4-PHASES

Figure 5: **2D projections of the Fashion MNIST dataset using UMATO and UMATO with multi-phase optimizations.** Although there was a small difference such as the locations of outliers, we observed that the projection results were quite similar to each other.

## D  PROJECTION STABILITY

Table 6 denotes the results of our experiment on the projection stability of UMATO and other dimensionality reduction techniques. When the data size grows, we want to sample a portion of it to speed up the visualization. However, the concern is whether the projection run with the sampled indices is consistent with the part of the projection result with indices selected from the full dataset. If the algorithm can generate stable and consistent results, the two projections should contain the least bias possible. To compute the projection stability of dimensionality reduction techniques, we

used the normalized Procrustes distance (Equation 8) to measure the distance between two comparable distributions. To replicate the experiment by McInnes et al. (2018), we used the same Flow Cytometry dataset (Spidlen et al. (2012); Brodie et al. (2013)), and ran optimal translation, uniform scaling, and rotation to minimize the Procrustes distance between the two distributions. As we can see in Table 6, UMATO outperformed $t$-SNE and A$t$-SNE for all sub-sample sizes. Moreover, although UMAP is known as stable among existing algorithms, UMATO showed even better (lower) Procrustes distance except for one sub-sample size (60%). From this result, we can acknowledge that UMATO can generate the more stable and consistent results regardless of sub-sample size than many other dimensionality reduction techniques.

Table 6: **The normalized Procrustes distance between two projection results by the percentage of sub-samples.** From four dimensionality reduction techniques, we measured the normalized Procrustes distance to check the projection stability using the Flow Cytometry dataset. The winner is in bold.

| Sample (%) | 1 | 2 | 5 | 10 | 20 | 30 | 50 | 60 | 80 | 100 |
|---|---|---|---|---|---|---|---|---|---|---|
| $t$-SNE | 0.9835 | 0.9944 | 0.9544 | 0.9736 | 0.9824 | 0.9924 | 0.9959 | 0.9819 | 0.9944 | 0.9765 |
| UMAP | 0.4002 | 0.3319 | 0.2341 | 0.1324 | 0.1327 | 0.1577 | 0.1109 | **0.0713** | 0.0951 | 0.0597 |
| A$t$-SNE | 0.8958 | 0.9510 | 0.7593 | 0.7980 | 0.9062 | 0.9376 | 0.9999 | 0.9460 | 0.9599 | 0.9999 |
| UMATO | **0.3153** | **0.1528** | **0.1206** | **0.0988** | **0.0520** | **0.1411** | **0.0526** | 0.1732 | **0.0529** | **0.0535** |

# E  QUALITY METRICS

As UMATO presents a dimensionality reduction technique that can capture both the global and local structures of high-dimensional data, we used several quality metrics to evaluate each aspect respectively. We have referred to some review papers (Gracia et al. (2014); Lee & Verleysen (2009)) for the best use and implementation. Among many quality metrics, we leveraged 1) Distance To a Measure (DTM, Chazal et al. (2011; 2017)), 2) KL divergence between two density functions, 3) trustworthiness and continuity (Venna & Kaski (2001)), and 4) mean relative rank errors (MRREs, Lee & Verleysen (2007)). The first two metrics are used to test the preservation of global structures and the last two metrics are suggested for the preservation of the local structures.

Distance To a Measure considers the dispersion of high- and low-dimensional data, where it is defined for a given point as $f_\sigma^X(x) := \sum_{y \in X} \exp\left(-\mathrm{dist}(x, y)^2/\sigma\right)$. By summing up the element-wise absolute values between two distributions, $\sum_{x \in X, z \in Z} f_\sigma^X(x) - f_\sigma^Z(z)$ where $x$ is the point in high-dimensional space $X$ and the $z$ is the corresponding projected point in low-dimensional space $Z$, we can examines the similarity of two datasets. In our experiments, we used the Euclidean distance and the values were normalized between 0 and 1. The $\sigma \in \mathcal{R}_{>0}$, which represents the length scale parameter, was set to 0.1. Moor et al. (2020) proposed the KL divergence of two probability distributions, $\mathrm{KL}_\sigma := \mathrm{KL}(f_\sigma^X || f_\sigma^Z)$, a variation of DTM. Changing $\sigma$ as a normalizing factor of the distribution, the authors investigated if the algorithms can preserve the global structure of the high-dimensional data. Following the same notion as the experiment in the paper (Moor et al. (2020)), we used three $\sigma$ values, 1.0, 0.1, and 0.01, to test whether each algorithm can capture the global aspect with respect to diverse density estimates.

Trustworthiness and continuity measure how much the nearest neighbors are preserved in a space (i.e., high- or low-dimensional space) compared to the other space by analyzing the ranks of $k$-nearest neighbors in both spaces. The difference between trustworthiness and continuity comes from which space is held as the base space. Specifically, we first need to find the $k$-nearest neighbors in both high- and low-dimensional space. Then, we compute the trustworthiness by checking whether the ranks of nearest neighbors in low-dimensional space resemble those of high-dimensional space. If so, we can achieve a high score in trustworthiness. Meanwhile, we achieve a high score in continuity if the ranks of nearest neighbors in high-dimensional space resemble those of low-dimensional space. MRREs take a similar approach to trustworthiness and continuity as it calculates and compares the ranks of the $k$-nearest neighbors in both spaces, but the normalizing factor is slightly different. Originally, it was better if MRREs had lower values. However, for the ease of comparing

local quality metrics, we defined it as MRREs := $1 -$ MRREs, so higher MRREs mean that they have better retained the $k$-nearest neighbors like trustworthiness and continuity.

## F  HYPERPARAMETER SETTING

As explained in Section 5.2, we generated projections for each dimensionality reduction algorithm that had the lowest $KL_{0.1}$ measure. To tune each algorithm's hyperparameters, we employed the grid search for $t$-SNE, UMAP, and A$t$-SNE. For $t$-SNE and A$t$-SNE, we changed the perplexity from 5 to 50 with an interval of 5, and the learning rate from 0.1 to 1.0 with a log-uniform scale. In the case of UMAP, we changed the number of nearest neighbors from 5 to 50 with an interval of 5, and the minimum distance between points in the projection from 0.1 to 1.0 with an interval of 0.1. We used the Python library scikit-optimize (Head et al. (2018)) to find the best hyperparameters for topological autoencoders. UMATO has several hyperparameters such as the number of hub points, the number of epochs, and the learning rate for global and local optimization. In our experiments, we configured everything except the number of hub points to the same setting for UMATO. We used 200 hub points for the Spheres dataset and had 300 hubs for others. We used fewer hub points for the Spheres since it has only 10,000 data points in total, while the other datasets have 60,000 data points. We set the number of epochs to 100 for global optimization and to 50 for local optimization. Lastly, the global learning rate was set to 0.0065, and the local learning rate was set to 0.01.

## G  SYNTHETIC SPHERES DATASET

We leveraged the same Spheres dataset that Moor et al. (2020) used in their experiments of topological antoencoders. The Spheres dataset contains eleven high-dimensional spheres which reside in 101-dimensional space. We first generated ten spheres of radius of 5, and shifted each sphere by adding the same Gaussian noise to a random direction. For this aim, we created $d$-dimensional Gaussian vectors $X \sim N(0, I(10/\sqrt{d}))$, where $d$ is 101. As to embed an interesting geometrical structure to the dataset, the ten spheres of relatively small radii of 5 were enclosed by another larger sphere of radius of 25.

## H  MORE EXPERIMENTS ON SYNTHETIC DATASETS

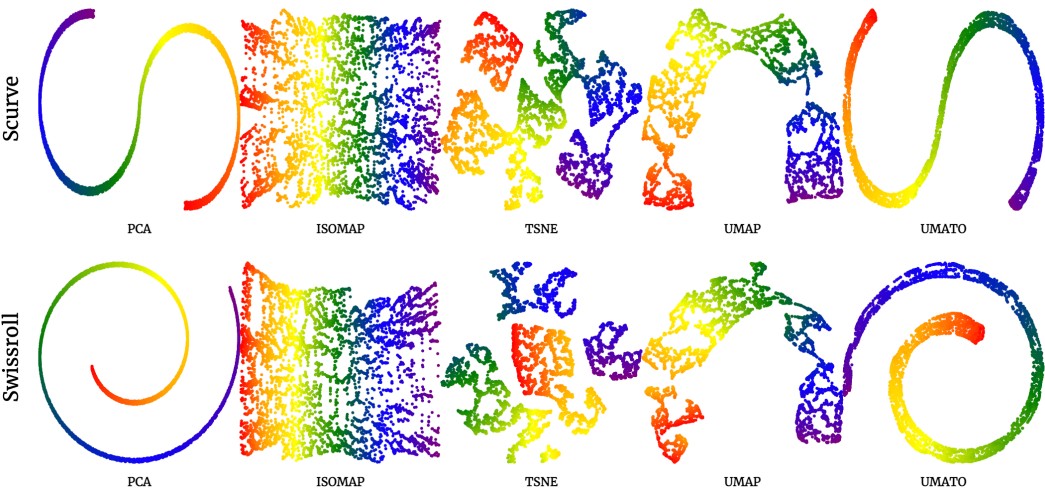

Figure 6: **2D projections produced by UMATO and four baseline algorithms.** 3-dimensional S-curve and Swiss roll datasets are used for five different algorithms. While PCA and UMATO can capture both the global and local structures of original datasets, other algorithms such as Isomap, $t$-SNE, and UMAP can only preserve the local manifolds of original datasets.

We leveraged the 3-dimensional S-curve and Swiss roll datasets to test whether UMATO can preserve both the global and local structures of original datasets. As the visualization shows (Figure 6),

only PCA and UMATO were able to capture the global and local structures of original datasets. Isomap, $t$-SNE and UMAP could capture the local manifolds of original datasets, but high-level manifolds of the original datasets were not reflected to the embedding.

# I   LOCAL QUALITY METRICS WITH DIFFERENT VALUES OF HYPERPARAMETER

We report the result of local quality metrics with diverse hyperparameters (Table 7). We changed the number of nearest neighbors ($k$) from 5 to 15 with an interval of 5. As we have already reported when the $k = 5$ (Table 2), below are the cases where $k = 10, 15$. As we can check from the result, while the values fluctuate a little bit, the ranks are mostly robust over diverse $k$ values.

Table 7: **Local quality metrics of UMATO and the baseline algorithms.** Although the values are changing a little bit depending on the number of nearest neighbors, when comparing the result of $k = 10$ and $k = 15$, the ranks barely change. The winner is in bold and underlined, and the runner-up in bold.

| Dataset | Method | Cont | Trust | MRRE$_X$ | MRRE$_Z$ | Dataset | Method | Cont | Trust | MRRE$_X$ | MRRE$_Z$ |
|---|---|---|---|---|---|---|---|---|---|---|---|
| Spheres $k = 10$ | PCA | 0.7965 | 0.6111 | 0.7976 | 0.6089 | MNIST $k = 10$ | PCA | 0.9519 | 0.7341 | 0.9574 | 0.7342 |
| | Isomap | **0.8847** | 0.6268 | **0.8953** | 0.6267 | | Isomap | 0.9713 | 0.7531 | 0.9743 | 0.7530 |
| | $t$-SNE | **0.8752** | **0.6803** | 0.8944 | **0.7082** | | $t$-SNE | **0.9779** | **0.9930** | 0.9838 | **0.9951** |
| | UMAP | 0.8688 | 0.6508 | 0.8764 | 0.6497 | | UMAP | **0.9858** | 0.9543 | **0.9889** | 0.9545 |
| | TopoAE | 0.8309 | 0.6354 | 0.8312 | 0.6335 | | TopoAE | 0.9686 | 0.9429 | 0.9716 | 0.9429 |
| | A$t$-SNE | 0.8645 | 0.6453 | 0.8724 | 0.6434 | | A$t$-SNE | 0.9688 | 0.9734 | 0.9782 | 0.9761 |
| | UMATO | 0.7875 | **0.6564** | 0.7881 | **0.6559** | | UMATO | 0.9753 | 0.8422 | 0.9792 | 0.8422 |
| Spheres $k = 15$ | PCA | 0.7952 | 0.6120 | 0.7969 | 0.6094 | MNIST $k = 15$ | PCA | 0.9481 | 0.7341 | 0.9555 | 0.7342 |
| | Isomap | **0.8774** | 0.6282 | **0.8914** | 0.6271 | | Isomap | 0.9692 | 0.7529 | 0.9732 | 0.7529 |
| | $t$-SNE | **0.8668** | **0.6723** | 0.8891 | **0.7021** | | $t$-SNE | **0.9746** | **0.9908** | 0.9819 | **0.9940** |
| | UMAP | 0.8639 | 0.6534 | 0.8737 | 0.6506 | | UMAP | **0.9834** | 0.9542 | **0.9877** | 0.9545 |
| | TopoAE | 0.8304 | 0.6364 | 0.8309 | 0.6339 | | TopoAE | 0.9666 | 0.9429 | 0.9705 | 0.9429 |
| | A$t$-SNE | 0.8603 | 0.6461 | 0.8699 | 0.6438 | | A$t$-SNE | 0.9642 | 0.9707 | 0.9755 | 0.9749 |
| | UMATO | 0.7875 | **0.6568** | 0.7879 | **0.6560** | | UMATO | 0.9728 | 0.8417 | 0.9778 | 0.8420 |
| Fashion MNIST $k = 10$ | PCA | 0.9827 | 0.9120 | 0.9843 | 0.9116 | Kuzushiji MNIST $k = 10$ | PCA | 0.9313 | 0.7218 | 0.9382 | 0.7214 |
| | Isomap | 0.9854 | 0.9196 | 0.9865 | 0.9196 | | Isomap | 0.9546 | 0.7639 | 0.9574 | 0.7636 |
| | $t$-SNE | 0.9873 | **0.9936** | 0.9903 | **0.9948** | | $t$-SNE | 0.9794 | **0.9677** | 0.9844 | **0.9687** |
| | UMAP | **0.9941** | 0.9586 | **0.9950** | 0.9585 | | UMAP | **0.9861** | 0.9556 | **0.9891** | 0.9561 |
| | TopoAE | **0.9898** | 0.9591 | 0.9908 | 0.9590 | | TopoAE | 0.9732 | 0.9442 | 0.9755 | 0.9441 |
| | A$t$-SNE | 0.9792 | **0.9843** | 0.9829 | **0.9846** | | A$t$-SNE | 0.9723 | **0.9659** | 0.9789 | **0.9670** |
| | UMATO | 0.9897 | 0.9498 | **0.9911** | 0.9500 | | UMATO | **0.9836** | 0.8880 | **0.9864** | 0.8890 |
| Fashion MNIST $k = 15$ | PCA | 0.9815 | 0.9121 | 0.9837 | 0.9117 | Kuzushiji MNIST $k = 15$ | PCA | 0.9266 | 0.7220 | 0.9358 | 0.7215 |
| | Isomap | 0.9838 | 0.9197 | 0.9858 | 0.9196 | | Isomap | 0.9497 | 0.7640 | 0.9553 | 0.7636 |
| | $t$-SNE | 0.9858 | **0.9894** | 0.9944 | **0.9927** | | $t$-SNE | 0.9758 | **0.9668** | 0.9825 | **0.9683** |
| | UMAP | **0.9934** | 0.9585 | **0.9946** | 0.9584 | | UMAP | **0.9836** | 0.9551 | **0.9879** | 0.9558 |
| | TopoAE | **0.9892** | 0.9591 | 0.9904 | 0.9590 | | TopoAE | 0.9715 | 0.9442 | 0.9746 | 0.9441 |
| | A$t$-SNE | 0.9769 | **0.9840** | 0.9817 | **0.9845** | | A$t$-SNE | 0.9678 | **0.9648** | 0.9765 | **0.9665** |
| | UMATO | 0.9888 | 0.9495 | 0.9906 | 0.9499 | | UMATO | **0.9814** | 0.8876 | **0.9853** | 0.8887 |

# J   2D EMBEDDING RESULTS OF UMATO AND BASELINE ALGORITHMS ON REAL-WORLD DATASETS

For the real-world datasets, UMATO showed a similar projection to PCA but with better captured local characteristics. The results from topological autoencoders showed some points detached far apart from their centers, even though the best hyperparameters were used for each. Although At-SNE claimed that it could capture both structures, the results were not significantly different from those of the original t-SNE algorithm when projecting the Spheres and Fashion MNIST datasets.

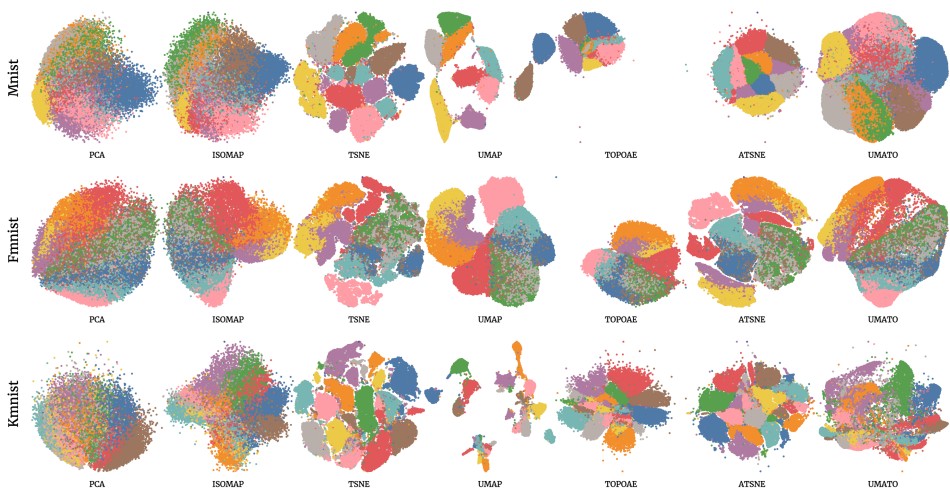

Figure 7: **2D projections produced by UMATO and six baseline algorithms** UMATO generated similar projections to PCA but with the points more locally connected; this is best viewed in color.

## K  EMBEDDING ROBUSTNESS OVER NUMBER OF EPOCHS

We report the experimental result in Figure 8. As we explained, the UMAP embedding results are susceptible to the number of epochs so that the distance between clusters get dispersed. This can induce a misinterpretation that the user considers the distance between clusters as something meaningful. The two-phase optimization of UMATO can solve the problem since the global optimization (first phase) is easy to converge as it runs only with a small portion of points. Therefore, the increasing number of epochs in the global optimization does not harm the final embedding.

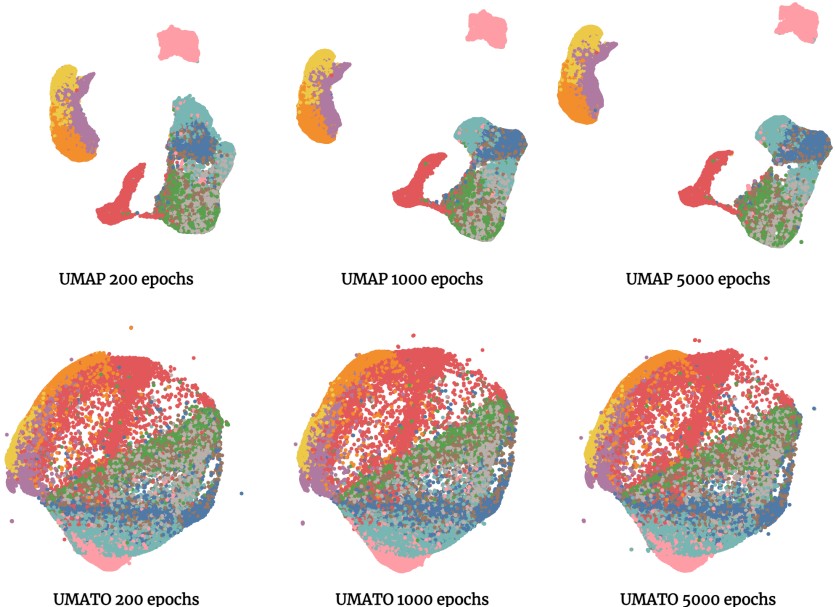

Figure 8: **Comparing result of UMATO and UMAP with varying number of epochs** (Top row) UMAP is susceptible to the number of epochs so that the clusters get dispersed as the epochs increases. (Bottom row) On the other hand, regardless of the number of epochs in the global optimization, UMATO results in almost the same embedding result.

## L    ILLUSTRATION OF UMATO PIPELINE

For the ease of understanding, we provide an illustration of UMATO pipeline in Figure 9. The detailed explanation for UMATO can be found in Section 4.

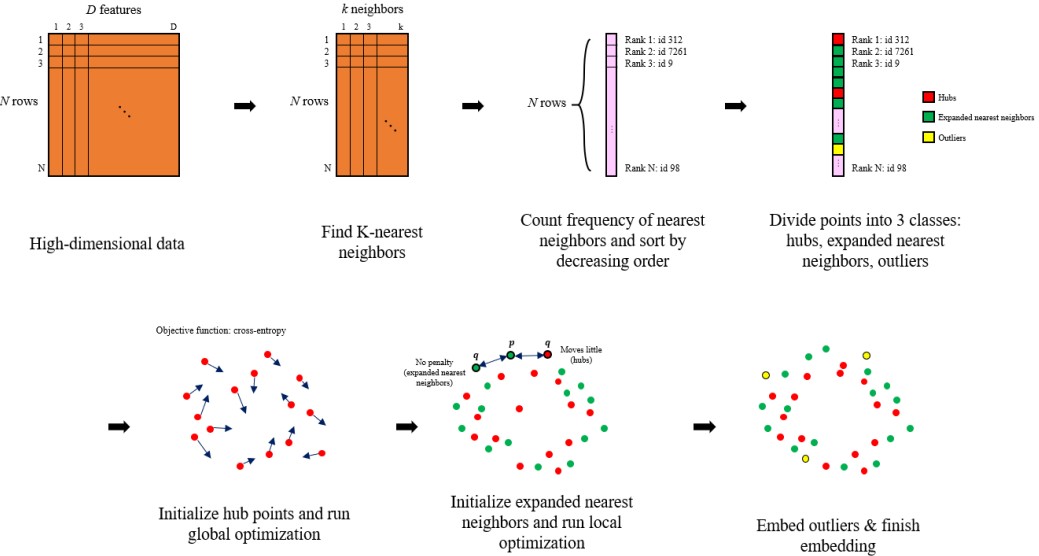

Figure 9: **The illustration of overall UMATO pipeline.**

## M    EFFECT OF LOCAL LEARNING RATE OF UMATO

By manipulating one of UMATO's hyperparameters, `local_learning_rate`, the user can determine where to focus in the embedding result; to reveal more of the global structures, the user should apply a lower value (e.g., 0.005), while using a higher one (e.g., 0.1) would generate more like a UMAP embedding which prefers to show the local manifolds.

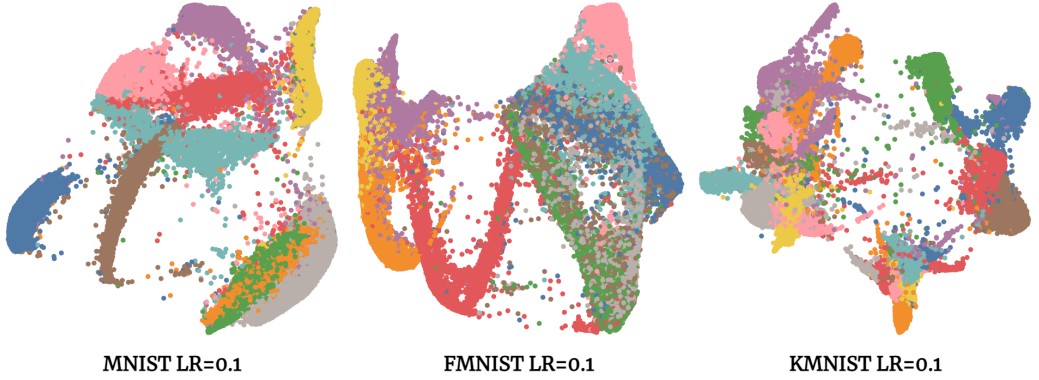

Figure 10: **The effect of manipulating local learning rate of UMATO.** Local learning rate was set to 0.1 for all cases. Unlike previous embedding results in Figure 7, UMATO reveals more of the local aspects.

## N    UMATO ON REAL-WORLD BIOLOGICAL DATASET

To test UMATO on the real-world biological dataset, we took professional advice from an expert who has a Ph.D. in Bioinformatics. We have run UMATO and the baseline algorithms (t-SNE, UMAP) on 23,822 single-cell transcriptomes from two areas at distant poles of the mouse neocortex (Tasic et al. (2018)). Each cell belongs to one of 133 clusters defined by Jaccard–Louvain clustering (for more than 4,000 cells) or a combination of k-means and Ward's hierarchical clustering. Likewise, each cluster belongs to one of 4 classes: GABAergic (red/purple), Endothelial (brown), Glutamatergic (blue/green), Non-Neuronal (dark green).

The embedding result for each method is given in Figure 11. In the case of t-SNE, clusters are well-captured, but the classes are much dispersed, while UMAP adequately separates both classes and clusters. Compared to these baseline algorithms, UMATO is able to capture the relationship between classes much better, retaining some of the local manifolds as well. This means that UMATO focuses more on the manifold at a higher level than the baselines that the hub points worked as the representatives that explain well about the overall dataset. Moreover, there are cases in biological data analysis where the researchers want to know the distance between samples González-Blas et al. (2019); Van den Berge et al. (2020). As the UMAP embedding results are susceptible to the number of epochs, this may cause a negative impact to interpret the results accurately. On the other hand, as UMATO is robust over the number of epochs, we do not have to worry about such biases.

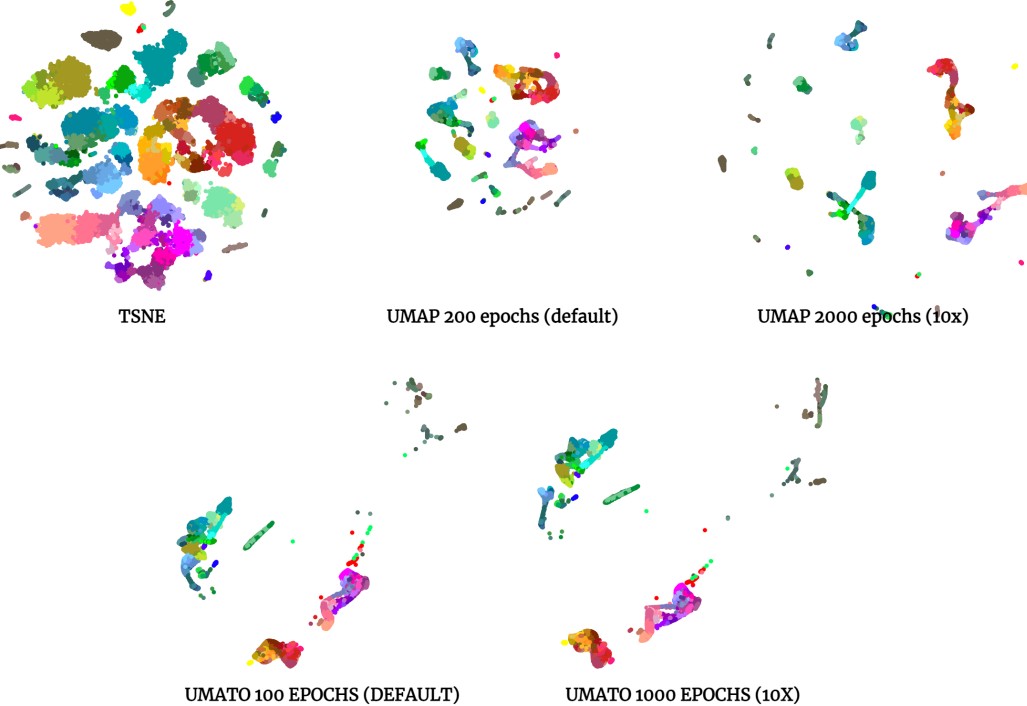

Figure 11: **2D visualization of UMATO, UMAP and *t*-SNE on mouse neocortex dataset (Tasic et al. (2018)).** *t*-SNE separates clusters well but does not show class information: GABAergic (red/purple), Endothelial (brown), Glutamatergic (blue/green), Non-Neuronal (dark green). UMAP moderately captures both the clusters and classes. In the case of UMATO, it demonstrates the relationship between classes much better than *t*-SNE and UMAP, retaining some of the local manifolds as well.

