# OpenReview forum: "Uniform Manifold Approximation with Two-phase Optimization"
_ICLR.cc/2021/Conference — Reject_

### Official Review · AnonReviewer1 · 2020-10-23
**Manifold embedding technique for clustered data relying on UMAP**

**Rating:** 6
**Confidence:** 4

**Review:**

The paper proposes UMATO, a low-dimensional embedding technique based on a two-phase optimization, the first phase embeds representative points (called hubs) for global structure preservation, and the second phase embeds the rest of the points for local structure preservation.

Quality, clarity, originality and significance:
Pro: The paper is well written and most aspects are clearly explained. It seems to me that the method is mostly designed for clustered data which is significant in many real world applications, but this is not emphasized in the paper, see also comments below. Overall, I found the paper clear, and I like that the authors show the connection to UMAP. Cons: As discussed below I find that there are some details missing, that have the potential to better explain why and how the method works. As the method borrows ideas from other methods, clearly discussing all the details is very important in order to increase its strengths.

Due to the two-phase approach to the embedding it seems that the method is much more adapted to clustered data than to the standard manifold embedding. How would the method perform for a continuous manifold that is fairly uniformly sampled such as the Swiss roll, would it create holes because of the two-phase structure? The results also show that UMATO performs best in the case of the Spheres dataset where the clusters are clearly separated. Could the authors extend on this?

Isomap has proven to be fairly good at embedding both the global and local structure of the data, I believe a comparison would be relevant and needed. More so, since from Table 2 it seems that UMATO is better at capturing the global rather than the local structure which is in line with what Isomap (not Landmark Isomap) is trying to do.

Would it be possible to define a score that combines the global structure and local structure preservation? For comparing the different methods that would be easier. What are the parameters of the different quality metrics used? The local quality metrics probably depend on a number of nearest neighbours. How was this chosen and are the results robust to the choice of parameters?

As UMATO follows the same line as UMAP, could the authors explain why did they choose to use a different initialization (PCA) and not the initialization in UMAP (spectral embedding)? What happens actually for different initializations? Does the stability of UMATO discussed in the paper depend on the stability of PCA or is independent of it? If PCA does not work, then the hubs will be wrongly embedded and the global structure will suffer.

Sect. 4.1.: “the most popular point … but not too densely connected” – How is the density estimated and controlled?; “we unfold it” – what does “it” refer to?

Sect. 4.2: p and q refer to points but the notation is often used for probabilities. A different notation might make things easier to read.

Sect. 4.4.: “we arrange $x_i$ using the position of $x_j$” – I might be missing smth, but should it be $y_i$ and $y_j$?

Sect. 5: How many connected components are identified in each of the experiments?

-----------------------------

Rebuttal: Thank you to the authors for addressing the comments and for the changes to the paper, and thank you for adding the examples on Scurve and Swiss Roll and the comparison with Isomap. I find it slightly surprising that Isomap performs very well for local metrics on the spheres dataset, because Isomap tends to preserve larger distances. Not sure I fully understand why.

Related to my initial concern about clustered data, my impression would be that in addition to the spheres visual example in the main paper, it would have been good to add the Scurve or the Swiss roll to emphasize that UMATO is not specifically designed for clustered data (all the examples in Table 2 are for data with implicit clusters). How do the methods perform in terms of the local and global metrics for the Scurve and Swiss roll datasets? Would zooming in on PCA reveal similar local structures to UMATO?

What is the difference between y_i and y_i' used in eq (8)? Are they the same?

Second line on page 2: Should it be UMATO instead of UMAP?

---

> ### Author Response · Authors · 2020-11-24
> **Response to Reviewer #1 (2/2)**
>
> **Q6-1. In Sect. 4.1: How is the density estimated and controlled?**
>
> A6-1. We can manipulate the density using the number of nearest neighbors (k) for each hub point. We can consider this a local manifold from which k points are sampled. Larger k suggests dense sampling, while smaller k does sparse sampling.
>
> ---
>
> **Q6-2. In Sect. 4.1: “we unfold it” – what does “it” refer to?**
>
> A6-2. "it" refers to the k-nearest neighbors indices. For example, if we have 10,000 points in total and 5 nearest neighbors for each point, we will have 50,000 indices in total as a 2-dimensional matrix (10,000 * 5). We unfold it as a 1-dimensional matrix (50,000 * 1) and count the frequency of each point so we can see the popularity of each point. We clarified that sentence.
>
> ---
>
> **Q7. In Sect. 4.2: change the notation (p and q)**
>
> A7. We fixed the notation (p -> f, q -> g). Thank you for the thorough feedback.
>
> ---
>
> **Q8. In Sect. 4.4: x_i should be y_i**
>
> A8. That was a typo and we have fixed it.
>
> ---
>
> **Q9. In Sect. 5: How many connected components are identified in each of the experiments?**
>
> A9. The term “connected component” we defined and used in Section 4 refers to a hub point and its neighbors. Therefore, the number of connected components is the same as the number of hub points. The number of hubs (connected components) was 200 for the Spheres dataset and 300 for the other datasets. The reason that we used more hub points was that Spheres had 10,000 points in total while others had 60,000 in total.
>
>
> ---
> Please let us know if there are any concerns or questions.
>
> Thanks again for your valuable time for reviewing our paper.
>
> Best,
>
> Authors
>
> [1] Michael Moor, Max Horn, Bastian Rieck, and Karsten Borgwardt. Topological autoencoders. In Proceedings of the 37th International Conference on Machine Learning (ICML), Proceedings of Machine Learning Research. PMLR, 2020.
>
> [2] https://github.com/BorgwardtLab/topological-autoencoders/blob/37fe9b2fc77161f0952eeea0a81d4b220575b120/test/test_measures.py#L19
>
> [3] Tenenbaum, J. B., De Silva, V., & Langford, J. C. (2000). A global geometric framework for nonlinear dimensionality reduction. science, 290(5500), 2319-2323.

---

> ### Author Response · Authors · 2020-11-24
> **Response to Reviewer #1 (1/2)**
>
> Dear reviewer 1,
>
> Thanks for your constructive feedback. We address your questions and concerns below.
>
> ---
>
> **Q1. Extend UMATO for a continuous manifold that is fairly uniformly sampled such as the Swiss roll.**
>
> A1. We have experimented UMATO on synthetic data such as Swiss roll and S curve which have uniformly sampled points constituting continuous manifolds. We also report the visualization results of the baseline algorithms such as PCA, Isomap, t-SNE, and UMAP (Appendix H). The visualization results show that UMATO reveals a clear description of the original high-dimensional global and local structures on both the S curve and Swiss roll. On the other hand, other algorithms such as Isomap, t-SNE, and UMAP reflect only the local manifold (See Figure 6 in Appendix H).
>
> ---
>
> **Q2. Add Isomap to the baseline and report the comparing result.**
>
> A2. We have added Isomap [3] to the baseline. We found that although Isomap has shown some performance with respect to global quality metrics, UMATO has still outperformed all the baseline algorithms in terms of KL_{0.1} in Spheres, Fashion MNIST and Kuzushiji MNIST. UMATO was the runner-up in the MNIST dataset. Moreover, UMATO was the only method that could show good performance in both global and local quality metrics. We also report that Isomap took more than 3 hours on average to make the embedding result on the MNIST dataset, while UMATO took about 73 seconds (Table 1).
>
> ---
>
> **Q3. Make a score that combines both local and global structures**
>
> A3. We are glad that you pointed this out. We also think that defining a score that can combine both the global and local structures could make the evaluation clearer. Also, we agree with the necessity of such a metric. However, we think defining a metric could bring about a bias that it could favor our work. Therefore, we adopted existing metrics for evaluation.
>
> ---
>
> **Q4-1. What are the parameters of the different quality metrics used?**
>
> A4-1. (global metrics) In the case of global quality metrics, sigma (i.e., the density over data) was the only hyperparameter. We tested three different values (i.e., 1.0, 0.1, 0.01) for the evaluation.
>
> ---
>
> **Q4-2. The local quality metrics probably depend on a number of nearest neighbours. How was this chosen and are the results robust to the choice of parameters?**
>
> A4-2. (local metrics) You are exactly right. We have set the number of nearest neighbors($k$) as 5 and chosen this following the default setting of one of our baseline algorithms, topological autoencoders [1, 2]. Furthermore, we report the result of local quality metrics using two additional $k$ values (10 and 15) in Appendix I (Table 7). We found that different $k$s barely affect the performance ranks of UMATO.
>
> ---
>
> **Q5. Why use PCA and not the spectral embedding? What happens actually for different initializations? Does the stability of UMATO discussed in the paper depend on the stability of PCA or is independent of it?**
>
> A5. For UMATO, we have tested several initialization methods such as spectral embedding, random, PCA, and class-wise separation where the results were almost the same on the real-world datasets. On the other hand, the UMAP and t-SNE results depend highly on the initialization method. We report this in Table 3 in Section 5.5 with a quantitative comparison using Procrustes distance we used for calculating the stability of UMATO embeddings. In the case of the Spheres dataset, as defined in Appendix F, the clusters were equidistant from each other. The embeddings have to be different due to the limitation of the 2-dimensional space since there is no way to express this relationship. However, as we report in Figure 4, the global and local structures of the data are manifested with UMATO with all different initialization methods.
> We finally chose PCA as it is fast and works even with large datasets. The robustness comes from the global optimization as it runs with only hub points so that the best positions of the hub points are easily determined. To sum up, the stability of UMATO does not come from the deterministic property of PCA but the global optimization process that we employ.

---

### Official Review · AnonReviewer2 · 2020-10-28
**A marginally improved version of UMAP for dimension reduction with reasonable experimental results but also some concerns for the method formulation.**

**Rating:** 5
**Confidence:** 4

**Review:**

### Summary:

This work proposed a dimensionality reduction algorithm called Uniform Manifold Approximation with Two-phase Optimization (UMATO), which is an improved version of UMAP (Ref. [3] see below). UMATO has a two-phase optimization approach: global optimization to obtain the overall skeleton of data & local optimization to identify the local structures.

On synthetic and three real-world datasets, UMATO achieved comparable and sometimes better results, compared to baseline method PCA, and alternative non-linear methods such as t-SNE, UMAP, topological autoencoders and Anchor t-SNE.

### Strengths & Originality:

The idea of splitting the optimization from all points in UMAP (Ref. [3]) into 3 parts, and only performing global optimization for hubs (selected from the original data) is interesting and reasonable.

Outlier detection is explicitly included in this proposed UMATO, this is good & as robustness is an important aspect for a practical manifold learning algorithm.

In section 1 "At-SNE inherits the conventional limitations of t-SNE ...... KL divergence exerts little effect on distant points in a high-dimensional space .....". For distribution p and q, once p is fixed (i,e., the encoding from original space X), as cross entropy H(p,q) = H(p) + KL(p,q), so these 2 become the same for optimizing y (associated q) from the embedding space. From equation 4, or appendix B equation 9, the claimed cross entropy function (leveraged from UMAP) is provided, and this is not the standard cross entropy defined in information theory. In fact, equation 4 = 0 when p = q, so this is more similar to KL & not cross entropy. Despite the terminology is unclear, the key difference between eq. 4 and KL is real, and seems reasonable to use eq. 4 as the objective function in the global optimization step.

### Weakness:

The first step of this proposed UMATO method is the points classification, i.e., assign each point into one of the 3 categories: hubs, expanded nearest neighbors, and outliers. This is an interesting concept, and related to previous works such as Ref. [4], which utilize the manifold ranking method to detect outliers. The main concern for this part, i.e., section 4.1, is the lack of formal formulation. Can we have some sort of objective function for this points classification step? It seems the quality of this step is quite important for the overall performance of UMATO, and seems no clear evidence to support the robustness here, from the experimental results in section 5.

The local optimization for each point in the "expanded NN" is also kind of Ad-hoc. There are several choices for this local optimization after the global optimization step, not exactly sure what is the key advantage of this proposed algo in section 4.3. For example, we can also use local alignment techniques to embed "expanded nearest neighbors", based on nearby hubs.

Initialization: as mentioned in section 3 (page 3), Laplacian eigenmap is applied as the initialization step for UAMP. For UMATO, the points classification step can be also viewed as an initialization (plus PCA initialization in the UMATO global optimization stage). However, it is unclear what is the corresponding step of t-SNE in experiments (section 5.2 & Appendix E). If there is no careful consideration, then it seems kind of unfair to say t-SNE perform worse than UMAP or UMATO. As all these methods are non-convex in terms of optimization, it is important to address this part more clearly.

Appendix B, page 12 "The KL divergence imposes a big penalty when v_{ij} is small but w_{ij} is large (Table 3 b.)", this seems to be a typo, as KL imposes a big penalty for "large v + small w" from Table 3 or equation 8.

Similarly, page 12 "the second term of cross-entropy imposes a big penalty when v_{ij} is large but w_{ij} is small (Table 3 g.)", however, Table 3.g seems to be "small v + large w", maybe a typo again.

### Reference:
This submission cited related works in manifold learning & dimension reduction & visualization such as Ref. [3], however, feel several related works are missing, for example Ref. [1] and Ref. [4] listed here.

1. Laurens van der Maaten. Accelerating t-SNE using Tree-Based Algorithms. Journal of Machine Learning Research, 2014.

2. James Cook, Ilya Sutskever, Andriy Mnih and Geoffrey Hinton. Visualizing similarity data with a mixture of maps. AISTATS 2007.

3. Leland McInnes, John Healy, and James Melville. Umap: Uniform manifold approximation and projection for dimension reduction. arXiv preprint arXiv:1802.03426, 2018.

4. Dian Gong, Xuemei Zhao, Gérard G. Medioni, Robust Multiple Manifold Structure Learning. ICML 2012.

---

> ### Author Response · Authors · 2020-11-24
> **Response to Reviewer #2**
>
> Dear reviewer 2,
>
> First of all, thanks for your time for reviewing our paper. We address your questions and concerns below.
>
> ---
>
> **Q1-1. Current explanations on UMATO seem ad hoc. Provide the reasons why we should use this specific UMATO algorithm.**
>
> A1-1. The problem of previous approaches like t-SNE and UMAP is that they are susceptible to different initialization methods, as they generate quite different embedding results depending on the initialization. In the case of t-SNE, this happens because of the fundamental limitation that KL divergence has; there is a little penalty for distant points in high-dimensional space being close in low-dimensional space. Meanwhile, UMAP utilizes the approximation technique to reduce the computation time, which requires additional hyperparameters such as 1) repulsion strength, 2) negative sample rate, and so on. Although this makes the computation much faster, this induces a significant problem where each cluster in the embedding gets dispersed as the number of epochs increases (Appendix K, Figure 8). This problem can induce a severe interpretation error as the user can conclude that the distances between clusters mean something. UMAP tried to alleviate this by fixing the number of epochs to 200, which is ad hoc, and by applying learning rate decay. However, the setting must change for each initialization method. It is, in most cases, hard to find the best setting in practice.
> To solve the aforementioned problems, we avoid using approximation during the optimization process, which normally would result in greatly increased computational cost. Therefore, we first run optimization only with a small number of hub points that represent the data well (i.e., uniformly sampled). Finding the optimal projection for a small number of points using the cross-entropy function is relatively easy and robust, making the additional techniques employed in UMAP unnecessary. Furthermore, it is less sensitive to the initialization method used (Section 5.5). After capturing the overall skeleton of embedding, we gradually append the rest of the points in multiple phases. Although the same approximation technique as UMAP is used for these points, as we have already embedded the hub points and use them as anchors, the projections become more robust and unbiased. The gradual appendage of points can in fact be done in a single phase; we found additional phases (i.e., more than two) do not result in meaningful improvements in the performance but only in the increased computation time (Section 4.5). As a result, we chose to use only two phases in UMATO: global optimization to capture the global structures (i.e., the pairwise distances in a high-dimensional space) and local optimization to retain the local structures (i.e., the relationships between neighboring points in a high-dimensional space) of data.
>
> ---
>
> **Q1-2. There is no clear evidence to support the robustness here, from the experimental results in Section 5.**
>
> A1-2. We report the robustness of each embedding technique with a quantitative comparison using Procrustes distance we used for calculating the stability of UMATO embeddings (Section 5.5). In the case of the Spheres dataset, as defined in Appendix F, each cluster is equally distant from one another, the embeddings have to be different since the 2-dimensional space has no way to express high-dimensional clusters that are equidistant. However, as we report in Figure 4, the global and local structures of the data are manifested with UMATO regardless of initialization methods. Moreover, the outlier detection process in UMATO makes the overall process more robust.
>
> | | $t$-SNE | UMAP | UMATO |
> | :--- |:---|:---|:---|
> |Spheres|0.7878|**0.7726**|0.9503|
> |MNIST|0.8665|0.7767|**0.4808**|
> |FMNIST|0.8284|0.7793|**0.0120**|
> |KMNIST|0.8668|0.8213|**0.2037**|
>
> ---
>
> **Q2. Carefully consider the initialization step of t-SNE and redo the experiments.**
>
> A2. Thanks for pointing this out. We found relevant work [1] that PCA initialization could improve the embedding performance of t-SNE. Thereafter, we did the same experiment again for t-SNE using PCA initialization. The experimental result is updated now. The result shows that, although the global metrics of t-SNE got a little better after using PCA as the initialization of embedding, still UMATO outperformed t-SNE in terms of KL_{0.1} in all datasets.
>
> ---
>
> **Q3. Address typos in Appendix B and add more references**
>
> A3. We now have updated this in the manuscript. Thanks for pointing this out.
>
> ---
>
> Thanks again for your detailed feedback.
>
> Please let us know if there are any more concerns left unsolved.
>
> Best,
>
> Authors
>
> [1] Linderman, G. C., Rachh, M., Hoskins, J. G., Steinerberger, S., & Kluger, Y. (2019). Fast interpolation-based t-SNE for improved visualization of single-cell RNA-seq data. Nature methods, 16(3), 243-245.

---

### Official Review · AnonReviewer3 · 2020-10-28
**Interesting visualization, but the algorithm description can be clearer**

**Rating:** 5
**Confidence:** 4

**Review:**

Summary: This paper applies Uniform Manifold Approximation and Projection (UMAP) recursively to a high-dimensional dataset to preserve both global and local structures for visualization. The proposed algorithm has two levels for the recursion: first run UMAP on the selected hubs, and then run UMAP for the nearest neighbors for each hub. I vote for marginally reject because the overall contribution looks incremental and the algorithm description is not super clear.

Strengths:
The paper presents a substantially improved visualization of the synthetic Spheres dataset, where the original global structure is preserved in the visualization, as well as quantitative results on four datasets (Spheres plus 3 real-world datasets).

Weaknesses:
1. By looking at the table of the quantitative results (Table 2), it seems hard to draw a conclusion at first sight, and the description of the results in Section 5.3 is not helpful either. It may be better if it is clearly called out that t-SNE and UMAP are good at local quality metrics, topological autoencoder and At-SNE are good at global quality metrics, while UMATO is good at both global and local metrics.
2. The visualization in Fig. 3 is interesting but I am not sure what information is conveyed or what conclusion is drawn for the three non-synthetic datasets. If the authors decide to keep these figures, more interpretations will be helpful for readers.
3. The algorithm description can be clearer by using some illustrative examples, besides using the actual Spheres data in Fig. 1 and Fig. 2. For example, building a graph, sort the data points in descending order of connectivity, and selecting the hubs: these steps can be well explained with an illustration, while using actual data such as Fig. 1 and Fig. 2 would make readers guess what is happening.
4. In Section 4.3, there are many descriptive steps in words that are vague (e.g. the description of an edge with point $p$ and $q$), and I don't precisely know what they refer to without a picture.
5. In Fig. 2(C), it shows "LOCAL INIT" which looks like the state before running the local optimization, while (A) (B) (D) show the states after the corresponding steps, but the descriptions in the figure title haven't clarified their meanings. In particular, what do (B) and (C) refer to exactly?

Questions during rebuttal period:
Please address #2 and #5 in the weaknesses above.

---

> ### Author Response · Authors · 2020-11-24
> **Response to Reviewer #3**
>
> Dear reviewer 3,
>
> Thank you for your detailed and helpful feedback. We address your questions and concerns below.
>
> ---
>
> **Q1. By looking at the table of the quantitative results (Table 2), it seems hard to draw a conclusion at first sight, and the description of the results in Section 5.3 is not helpful either. It may be better if it is clearly called out that t-SNE and UMAP are good at local quality metrics, topological autoencoders and At-SNE are good at global quality metrics, while UMATO is good at both global and local metrics.**
>
> A1. As you suggested, we changed Section 5.3 so that the reader can see the advantage of each algorithm at first sight. We also added Isomap [1] to the baseline as R1 suggested. To summarize the result, Isomap and topological autoencoders were good at global quality metrics, while UMAP, t-SNE and At-SNE were good at local quality metrics. UMATO was the only method showing the performance both in global and local quality metrics compared to the baseline algorithms.
>
> ---
>
> **Q2. The visualization in Fig. 3 is interesting but I am not sure what information is conveyed or what conclusion is drawn for the three non-synthetic datasets. If the authors decide to keep these figures, more interpretations will be helpful for readers.**
>
> A2. Thanks for pointing this out. We now report the visualization result of only the Spheres dataset and moved the others to the appendix to make the manuscript focus more on our main contributions which are: 1) robustness over diverse initialization methods and projection stability (Section 5.5, Appendix D) and 2) no such biases that UMAP has since UMATO leverages two-phase optimization (Appendix K).
>
> ---
>
> **Q3 & 4. The algorithm description can be clearer by using some illustrative examples.**
>
> A3 & 4. We agree that it would be easier for the readers to understand the main concept of UMATO with an illustrative example. We added an illustration of the UMATO pipeline in Appendix L (Figure 9).
>
> ---
>
> **Q5. Clarify the comments in Figure 2. In particular, what do Figure2-(B) and Figure2-(C) refer to exactly?**
>
> A5. We agree that the comments in Figure 2 were not clear. We have clarified the meanings of the comments in Figure 2. Specifically, we changed each comment as follows:
>
> (A)   Global init -> hub points initialization
>
> (B)   Global opt -> after optimizing the positions of hub points
>
> (C)   Local init -> after embedding expanded nearest neighbors
>
> (D)   Final result -> final embedding result
>
> ---
>
> Please let us know if there are any more concerns.
>
> Best,
>
> Authors
>
> [1] Tenenbaum, J. B., De Silva, V., & Langford, J. C. (2000). A global geometric framework for nonlinear dimensionality reduction. science, 290(5500), 2319-2323.

---

### Official Review · AnonReviewer4 · 2020-10-28
**results are not convincing**

**Rating:** 4
**Confidence:** 5

**Review:**

The authors attempt to address an important problem of preserving both the global and local structures of  high-dimensional data by applying two-step optimization approach to UMAP. The paper is well written and this is an interesting direction, but unfortunately the results don't look convincing at all:

1. Results from Table 2 demonstrate that UMATO performs better on global quality metrics, but looks like it achieves it by reducing local quality metrics. Figure 3 doesn't really help in convincing that the method is useful - if I will remove the labels, I will only see a blob and no separate clusters. I don't think that preservation of global metrics at this price is useful. It would be much more interesting to see that the clusters are still preserved, but they are positions relative one to another also correspond to a global structure.
2. MNIST and Fashion MNIST are certainly more realistic than simulated data, but would be nice to see the results on some real world datasets. For example, the authors give examples from biology in the introduction. Maybe they could show the performance and advantages of their method on one of these datasets. For example, a list of datasets for inspiration could be found here: https://genomebiology.biomedcentral.com/articles/10.1186/s13059-020-02128-7
3. In real world applications, UMAP or tSNE are rarely used with random initializations: https://www.biorxiv.org/content/10.1101/2019.12.19.877522v1. Therefore, I think it would be more fair to compare the proposed method with non-random initializations used in practice.

---

> ### Author Response · Authors · 2020-11-24
> **Response to Reviewer #4 (2/2)**
>
> ---
>
> **Q2. Test UMATO on real-world biological datasets.**
>
> A2. To test UMATO on the real-world biological dataset, we took professional advice from an expert who has a Ph.D. in Bioinformatics. We have run UMATO and the baseline algorithms (t-SNE, UMAP) on 23,822 single-cell transcriptomes from two areas at distant poles of the mouse neocortex [1]. Each cell belongs to one of 133 clusters defined by Jaccard–Louvain clustering (for more than 4,000 cells) or a combination of k-means and Ward’s hierarchical clustering. Likewise, each cluster belongs to one of 4 classes: GABAergic (red/purple), Endothelial (brown), Glutamatergic (blue/green), Non-Neuronal (dark green).
> We report the visualization results in Figure 11. In the case of t-SNE, clusters are well-captured, but the classes are much dispersed, while UMAP adequately separates both classes and clusters. Compared to the baseline algorithms, UMATO is able to capture the relationship between classes much better, retaining some of the local manifolds as well. This means that UMATO focuses more on the manifold at a higher level than the baselines that the hub points worked as the representatives that explain well about the overall dataset. Moreover, there are cases in biological data analysis where the researchers want to know the distance between samples [2, 3]. As the UMAP embedding results are susceptible to the number of epochs, this may make a negative impact on interpreting the results accurately. On the other hand, as UMATO is robust over the number of epochs, we do not have to worry about such biases.
>
> ---
>
> **Q3. Consider initialization steps for t-SNE and UMAP and redo the experiments.**
>
> A3. Thanks for pointing this out. As UMAP uses spectral embedding to initialize points as a default setting, we used it in all the experiments, although, we have not explicitly written this in Section 5. Thus, we updated the manuscript so that the readers do not confuse this. On the other hand, we found relevant work [4] that PCA initialization could improve the embedding performance of t-SNE. Therefore, we chose to use PCA for initializing t-SNE. The experimental result is updated now. The result shows that, although the global metrics of t-SNE got a little better after using PCA as the initialization of embedding, still UMATO outperformed t-SNE in terms of KL_{0.1} in all datasets.
> Again, thank you for your time and helpful feedback on our paper.
>
> ---
>
> Best,
>
> Authors
>
>
> [1] Tasic, B., Yao, Z., Graybuck, L. T., Smith, K. A., Nguyen, T. N., Bertagnolli, D., ... & Penn, O. (2018). Shared and distinct transcriptomic cell types across neocortical areas. Nature, 563(7729), 72-78.
>
> [2] González-Blas, C. B., Minnoye, L., Papasokrati, D., Aibar, S., Hulselmans, G., Christiaens, V., ... & Aerts, S. (2019). cisTopic: cis-regulatory topic modeling on single-cell ATAC-seq data. Nature methods, 16(5), 397-400.
>
> [3] Van den Berge, K., De Bezieux, H. R., Street, K., Saelens, W., Cannoodt, R., Saeys, Y., ... & Clement, L. (2020). Trajectory-based differential expression analysis for single-cell sequencing data. Nature communications, 11(1), 1-13.
>
> [4] Linderman, G. C., Rachh, M., Hoskins, J. G., Steinerberger, S., & Kluger, Y. (2019). Fast interpolation-based t-SNE for improved visualization of single-cell RNA-seq data. Nature methods, 16(3), 243-245.

---

> ### Author Response · Authors · 2020-11-24
> **Response to Reviewer #4 (1/2)**
>
> Dear reviewer 4,
>
> Thanks for your time for reviewing our paper. Below is a summary of how we address your concerns.
>
> ---
>
> **Q1. Convince that UMATO visualization is useful.**
>
> A1. We provide additional visualizations in Appendix M (Figure 10) for the results of UMATO by the distance between clusters. These results were produced by manipulating one of the hyperparameters in UMATO, learning rate (0.01 -> 0.1) in the local optimization so that the user can change to suit one’s needs.
> Moreover, the problem of previous approaches like t-SNE and UMAP is that they are susceptible to different initialization methods, as they generate quite different embedding results. In the case of t-SNE, this happens because of the fundamental limitation that KL divergence has; there is a little penalty for distant points in high-dimensional space being close in low-dimensional space. Meanwhile, several small reasons are interacting in UMAP. First, UMAP utilizes the approximation technique to reduce the computation time, which requires additional hyperparameters such as 1) repulsion strength, 2) negative sample rate, and so on. Although this makes the computation much faster, this induces a significant problem where each cluster in the embedding gets dispersed as the number of epochs increases (Appendix K, Figure 8). This problem can induce a severe interpretation error as the user can conclude that the distances between clusters mean something. UMAP tried to alleviate this by fixing the number of epochs to 200, which is ad hoc, and by applying learning rate decay. However, the setting must change for each initialization method which is, in most cases, hard to find the best one in practice.
> To solve the aforementioned problems, we avoid using approximation during the optimization process, which normally would result in greatly increased computational cost. Therefore, we first run optimization only with a small number of hub points that represent the data well (i.e., uniformly sampled). Finding the optimal projection for a small number of points using the cross-entropy function is relatively easy and robust, making the additional techniques employed in UMAP unnecessary. Furthermore, it is less sensitive to the initialization method used (Section 5.5). After capturing the overall skeleton of embedding, we gradually append the rest of the points in multiple phases. Although the same approximation technique as UMAP is used for these points, as we have already embedded the hub points and use them as anchors, the projections become more robust and unbiased. The gradual appendage of points can in fact be done in a single phase; we found additional phases (i.e., more than two) do not result in meaningful improvements in the performance but only in the increased computation time (Section 4.5). As a result, we chose to use only two phases in UMATO: global optimization to capture the global structures (i.e., the pairwise distances in a high-dimensional space) and local optimization to retain the local structures (i.e., the relationships between neighboring points in a high-dimensional space) of data.
> We report the robustness of each embedding technique with a quantitative comparison using Procrustes distance we used for calculating the stability of UMATO embeddings. In the case of the Spheres dataset, as defined in Appendix F, each cluster is equally distant from one another, the embeddings have to be different since the 2-dimensional space has no way to express high-dimensional clusters that are equidistant. However, as we reported in Figure 4, the global and local structures of the data are manifested with UMATO regardless of initialization methods. Moreover, the outlier detection process in UMATO makes the overall process more robust.
>
> We summarize the comparing result in a table form as below:
>
> | | $t$-SNE       | UMAP | UMATO |
> |----- | ----- |----| ----- |
> |Initialization methods (Section 5.5) | susceptible      | susceptible | **robust** |
> | Number of epochs (Appendix K)| -     | induce biases  |   **robust** |

---

### Decision · Program_Chairs · 2021-01-07
**Final Decision**

**Decision:**

Reject

**Comment:**

Reviewers generally agree that the proposed method UMATO, a two-phase optimization dimensionality reduction algorithm based on UMAP, is interesting and has potential, and that the paper is well-written. However, there are several concerns with the current paper. In particular, R1 is not convinced by the performance of UMATO on real-world datasets compared with previous methods such as t-SNE (see the linked papers). Both R1 and R2 are concerned that given the 2-phase approach, UMATO might be much more adapted to clustered data than standard manifold embedding. They pointed out that in the Swiss  roll/S-curve examples, UMATO stays very close to PCA, which is used for initialization, instead of globally unfolding the manifold as Isomap. These issues should be clarified/explored further for a better understanding and/or improvement of the current work.